# Improved Water and Waste Management Practices Reduce Diarrhea Risk in Children under Age Five in Rural Tanzania: A Community-Based, Cross-Sectional Analysis

**DOI:** 10.3390/ijerph19074218

**Published:** 2022-04-01

**Authors:** Paul H. McClelland, Claire T. Kenney, Federico Palacardo, Nicholas L. S. Roberts, Nicholas Luhende, Jason Chua, Jennifer Huang, Priyanka Patel, Leonardo Albertini Sanchez, Won J. Kim, John Kwon, Paul J. Christos, Madelon L. Finkel

**Affiliations:** 1Department of Surgery, NewYork-Presbyterian Brooklyn Methodist Hospital, New York, NY 11215, USA; 2Department of Population Health Sciences, Weill Cornell Medicine, New York, NY 10065, USA; clk2008@med.cornell.edu (C.T.K.); fep4002@med.cornell.edu (F.P.); nlr4002@med.cornell.edu (N.L.S.R.); jac7024@med.cornell.edu (J.C.); jennifer.huang.jh2988@yale.edu (J.H.); ppatel38@student.nymc.edu (P.P.); lea4007@med.cornell.edu (L.A.S.); wok4002@med.cornell.edu (W.J.K.); jyk4001@med.cornell.edu (J.K.); pac2001@med.cornell.edu (P.J.C.); maf2011@med.cornell.edu (M.L.F.); 3ASMK Foundation, Shinyanga P.O. Box 350, Tanzania; luhende.n@gmail.com

**Keywords:** WASH, drinking water, diarrhea, children under five, prevention, hygiene, sanitation, rural, Tanzania, Sub-Saharan Africa

## Abstract

Diarrhea remains a significant cause of morbidity and mortality among children in developing countries. Water, sanitation, and hygiene practices (WASH) have demonstrated improved diarrhea-related outcomes but may have limited implementation in certain communities. This study analyzes the adoption and effect of WASH-based practices on diarrhea in children under age five in the rural Busiya chiefdom in northwestern Tanzania. In a cross-sectional analysis spanning July-September 2019, 779 households representing 1338 under-five children were surveyed. Among households, 250 (32.1%) reported at least one child with diarrhea over a two-week interval. Diarrhea prevalence in under-five children was 25.6%. In per-household and per-child analyses, the strongest protective factors against childhood diarrhea included dedicated drinking water storage (OR 0.25, 95% CI 0.18–0.36; *p* < 0.001), improved waste management (OR 0.37, 95% CI 0.27–0.51; *p* < 0.001), and separation of drinking water (OR 0.38, 95% CI 0.24–0.59; *p* < 0.001). Improved water sources were associated with decreased risk of childhood diarrhea in per-household analysis (OR 0.72, 95% CI 0.52–0.99, *p* = 0.04), but not per-child analysis (OR 0.83, 95% CI 0.65–1.05, *p* = 0.13). Diarrhea was widely treated (87.5%), mostly with antibiotics (44.0%) and oral rehydration solution (27.3%). Targeting water transportation, storage, and sanitation is key to reducing diarrhea in rural populations with limited water access.

## 1. Introduction

Diarrheal diseases disproportionately affect young children under five years of age and are responsible for substantial morbidity and mortality in this cohort, particularly in low- and middle-income countries (LMICs) [1]. Unsafe drinking water practices, poor sanitation, limited educational opportunities, lack of medical infrastructure, and rapid population growth contribute to the prevalence of waterborne illness in LMICs [2,3,4,5,6]. In its 2015 Millennium Development Goals (MDG), the United Nations called for a global effort to reduce childhood diarrhea rates to fewer than 1 death per 1000 live births worldwide by 2025 [7,8,9,10]. Since the majority of childhood diarrhea cases originate from contaminated drinking water and fecal–oral transmission [11,12], a variety of strategies targeting water procurement, local sanitation, childhood nutrition, and diarrhea treatment have been implemented in different locales to improve outcomes. These strategies vary from direct investments in local infrastructure [5,6,13] to educational campaigns advocating better practice among individuals [3,4,14,15,16,17] to trial implementations of new technologies [2,18,19,20].

In recent decades, portions of Sub-Saharan Africa in particular have seen a dramatic improvement in young childhood mortality due to these efforts. In Tanzania, for example, there has been an 89.0% decrease in diarrhea-specific mortality in children under five years of age between 1980 and 2015, correlating with expansion of public infrastructure and increased distribution of antidiarrheal treatments throughout the country [21]. Despite this progress, the diarrhea-specific under-five mortality rate in Tanzania is still more than double the global average (50.31 vs. 21.60 deaths per 100,000 live births in 2017), and access to clean water and sanitation remains inconsistent throughout the country, with both urban and rural areas affected [22]. Tanzania is also one of several countries that did not meet MDG-stipulated clean water goals in 2015, and the Tanzanian Ministry of Health estimates that approximately 43% of Tanzanian households nationwide still use unsafe water sources for domestic purposes, 76% use unimproved pit latrines for waste, and 14% still practice open defecation [23,24,25]. As a result, childhood diarrhea remains a significant public health problem in many Tanzanian communities, even among those who have undergone multiple interventions to address the issue. 

The purpose of this study is to better understand the effect of different diarrhea prevention strategies on under-five childhood diarrhea in the rural Tanzanian setting. To this end, the utilization and outcomes of various water, sanitation, and hygiene (WASH) interventions were quantified and analyzed in a single district in rural Tanzania, from simple at-home practices to more advanced infrastructural improvements. Household-based surveys documenting water use and sanitation were used to determine risk factors for under-five childhood diarrhea in this population, and methods of diarrhea treatment were also queried. Analyzing these trends may shed light on optimal pathways to reduce young childhood diarrhea in the rural Sub-Saharan African setting while minimizing cost and ensuring sustainability.

## 2. Materials and Methods

### 2.1. Study Area and Population

Shinyanga region in northwestern Tanzania, the location of this study, is one of the driest regions of Tanzania (see Figure 1A). The climate of Shinyanga is semi-arid, with rainfall following a typical wet–dry tropical pattern characteristic of Eastern Africa (wet season November–April, dry season May–October, 850 mm total annual rainfall) [26]. According to the 2012 national census, the total population of the region is 1,534,808, of whom 281,849 (18.4%) are children under five years of age [27]. Within the region, only 14.9% of households have access to a tap water source. Among households that rely on wells and springs for water, only 31.0% and 6.0% use protected versions of these sources, respectively. Approximately 55,000 households (21.2%) have no reliable access to water and rely on rainfall and other surface water sources exclusively [28].

The specific site of this study was the Talaga, Ukenyenge, and Mwaweja wards, located within the Kishapu district in eastern Shinyanga. These three wards have a combined population of 25,077 and represent roughly one-tenth the population of greater Kishapu, which has a population of 272,990 (50,504 under-five individuals, 18.5%) [27]. This area was selected because of its centralized population, community infrastructure, and previous experience with international and Tanzanian non-governmental organizations (NGOs). Moreover, these three wards comprise the central portion of the historic Busiya chiefdom, which acts as a cultural and commercial hub for the surrounding area (see Figure 1B). The people of central Busiya are predominantly ethnically Sukuma, and most individuals practice subsistence farming as their primary occupation. Nearby urban centers include Shinyanga town (population 161,391) and Mwanza to the north (population 2,772,509) [29].

### 2.2. Previous Interventions and Current Clean Water Practices

Within central Busiya, a cluster of four adjacent villages was chosen for analysis: Nhobola, Negezi, Ngunga, and Ubata. Each of these villages is administered by an elected council of 6–12 members, with a local clinic capable of basic outpatient medical treatments available on-site (e.g., oral rehydration solution and antibiotics). Electricity is scarce in these communities, and water is generally procured from decentralized sources such as wells/springs, surface water, or occasionally public taps. All four selected villages have had multiple years of experience working with various clean water and diarrhea prevention initiatives spearheaded by local NGOs, primarily focusing on WASH-related practices such as handwashing, separation/treatment of drinking water, and proper waste disposal [30]. These initiatives receive support from both the local government and Busiya chiefdom traditional leaders, who provide additional advocacy via promotional campaigns and fundraising events. Since 2016, clean water has become a regular discussion topic at the annual cultural festival in Busiya (Ukenyenge), which draws thousands of individuals from the surrounding villages and occurs during the Sabasaba national holiday. This agenda has led to the recent improvement of several water sources in central Busiya, including the sealing of multiple wells and springs with concrete or metal enclosures as well as the installation of dedicated filtration systems on certain portions of available piping. At the time of analysis, no improved water sources had a known history of contamination.

In 2017, one of the largest year-round water sources in Nhobola was improved with a decentralized solar-powered water treatment center (SunSpring^®^, Innovative Water Technologies, Rocky Ford, CO, USA), which was installed by a US-based organization (H2Opendoors/Rotary International, San Carlos, CA, USA) with logistical support from Busiya chiefdom leadership and local NGOs (ASMK Foundation, Shinyanga, Tanzania). This treatment center is a pump-fed, end-of-pipe, membrane-filtered, standalone water filtration system that relies on reusable ultrafiltration modules for water treatment. Filtration is accomplished via pressurized, outside–in flow through cylindrical 0.02 μm-pore membranes that filter 99.999% of bacteria, viruses, and colloids while removing turbidity and other common organic contaminants. Daily filtration capacity is roughly 5000 gallons of water per day from most water sources, with self-cleaning mechanisms and replacement modules on-site to ensure longevity. A bottling station was included at the time of installation, as well as dedicated drinking water containers. To date, similar units have been installed in dozens of countries worldwide and have routinely outperformed their 10-year projected lifespan [31].

### 2.3. Study Design and Variables

This study was a household-based, cross-sectional survey of clean water and sanitation practices in central Busiya, as well as the prevalence of diarrhea in children under five years of age. Organizational framework and resource allocation were based on prior local census data collected by the Busiya chiefdom leadership and local NGOs in 2012 and 2016, which included demographic information as well as family statistics such as the age and sex of members in each household. These data were then used to locate households with children under five years old for study. Given that all or nearly all households with under-five children in the study area were intended to be interviewed, it was estimated that this study cohort would exceed standard statistical sample size calculations and provide enough power to draw appropriate conclusions about the population in question.

A 27-question household survey was drafted to assess water usage and health outcomes in central Busiya. This questionnaire was based on the 2009 WHO/UNICEF Core Questions on Drinking-Water and Sanitation for Household Surveys guidelines [32], translated into Swahili by a professional interpreter and modified with several site-specific adjustments. Questions for which multiple simultaneous answers were possible were written as ranked multiple-answer multiple-choice questions, allowing respondents to rank up to three choices based on frequency of use in the household. These questions included drinking/non-drinking water sources, water transportation methods, point-of-use water treatment strategies, sewage system availability, and methods of child waste disposal. Other questions in which answers were mutually exclusive (e.g., “yes/no” questions) were used for topics such as separation of drinking water and handwashing. Further questions regarding locations, demographics, or numbers (e.g., village, age, sex, household members, time required to procure water, etc.) were answered directly via fill-in responses. 

To simplify analysis, survey questions containing multiple ranked answers were recategorized into binary variables summarizing WASH compliance according to WHO/UNICEF recommendations (see Figure 2) [30,32]. For water procurement, WASH-compliant methodologies included use of dedicated filtered plumbing, use of improved wells or springs, purchase of commercial bottled water products, and transportation of drinking water in dedicated covered plastic containers. For storage and treatment of drinking water at home, separation of drinking water from other water sources as well as use of several point-of-use water treatment strategies (e.g., boiling, bleach/chlorine sterilization, mesh/cloth filtration, dedicated water filter use, and ultraviolet sterilization) were likewise considered to be WASH-compliant. Finally, regarding household sanitation, WASH compliance was defined as handwashing with soap, use of dedicated sewage systems or improved pit latrines, and hygienic disposal of child waste. In all scenarios, households were considered to be adherent to WASH practices only if WASH-compliant modalities were followed exclusively, with no concurrent noncompliant practices. Households using both WASH-compliant and non-WASH-compliant methods for a given aspect of water usage or sanitation were considered the be non-WASH-compliant for the corresponding summary variable.

The primary outcome of this study was the incidence of young childhood diarrhea, defined per WHO/UNICEF guidelines as watery stools more than three times per day in the last two weeks in children under five years of age [33]. Other health outcomes queried in this survey included quality of diarrhea (categorized as mucoid, watery, bloody, or with ova/parasites), two-week local clinic visitation for diarrhea, and type of antidiarrheal treatment received, if any (e.g., antibiotics, antiparasitics, oral rehydration solution, or traditional remedies). As with WASH compliance statistics, all health-related outcomes were directly reported by heads of households (HOHs).

### 2.4. Survey Implementation and Ethical Clearance

Data collection was conducted between July and September 2019 by a combined US–Tanzanian research team, with site visits coordinated by a local NGO (ASMK Foundation, Shinyanga, Tanzania) in partnership with Busiya chiefdom leaders. US-based researchers held on-site trainings for NGO staff and NGO-affiliated volunteer research assistants, many of whom were teachers or other local professionals who had volunteered previously to collect census data in Busiya community surveys. All research assistants were required to attend and complete a formal training for this analysis, regardless of previous experience. Prior to survey administration, open meetings were held at the villages involved, during which researchers were present and available to explain the purpose of the study, reinforce the voluntary nature of the survey, and answer any questions.

Surveys were printed in hardcopy, and answers were obtained during scheduled in-person interviews between research assistants and pre-identified HOHs. All interviews were conducted verbally in Swahili, and all research assistants were native speakers. Verbal responses from HOHs were directly recorded by research assistants at the time of interview. Once completed, surveys with handwritten answers were collected in a central repository in Shinyanga town, at which point they were organized by village, de-identified, digitized, encrypted, and analyzed. Incomplete surveys and surveys lacking initial identifying information were excluded from the study.

Permission was obtained from the local government district executive officer (DEO), local village councils, and the Busiya chiefdom traditional leadership prior to conducting the survey. Signed informed consent was obtained from all HOHs during survey interviews. Ethical clearance was granted by the Weill Cornell Medicine Institutional Review Board.

### 2.5. Statistical Analysis

Continuous variables were described as means (standard deviation) or medians (interquartile range, IQR) and were compared via Student’s *t*-test or Mann–Whitney *U* test, respectively. Categorical variables were described as frequencies or percentages and were compared via χ^2^ test. To determine predictors of childhood diarrhea, two parallel analyses were performed: one with per-household childhood diarrhea as an endpoint, and one analyzing aggregate incidence of diarrhea among all children included in the study. For both analyses, univariable logistic regression was performed on all potential predictive variables. Statistically significant variables found within this univariable analysis were subsequently tested in multivariable analysis, except variables in which one choice predominated (>70% of total replies) or the number of missing values was high (>10%). A simple multivariable logistic regression model was used for the per-household analysis, and a multivariable generalized estimating equations (GEE) model was performed for the per-child analysis, assuming an exchangeable working correlation structure (i.e., to account for clustering of children within the household). All *p*-values were two-sided with a statistical significance evaluated at the 0.05 alpha level. Ninety-five percent confidence intervals (95% CI) for all reported odds ratio estimates were computed to assess the precision of the obtained estimates. All statistical analyses were performed using SAS statistical software (v9.4, SAS Institute, Cary, NC, USA).

## 3. Results

### 3.1. Sociodemographic Characteristics

A total of 849 households were identified for interview for this study, of which 837 (98.5%) provided survey responses. Of these, 58 surveys were excluded for incompleteness, leaving 779 (93.1%) households to be included in the analysis. By village, Nhobola had the largest number of households with complete responses (261/265 households, 98.5%), followed by Ubata (198/200 households, 98.0%), Negezi (164/175 households, 93.7%), and Ngunga (156/197 households, 79.2%).

Among responding households, 4702 individuals were included in the analysis, of whom 1338 (28.5%) were children under five years of age (see Table 1). The median number of household members was 6 (IQR 4–7), and the median number of children under age five per household was 2 (IQR 1–2). The median age of under-five children was 3 (IQR 2–4) years. These household demographics were largely homogeneous among all four villages. Similarly, 95.8% of responding HOHs listed “subsistence farmer” as their primary occupation; this did not vary substantially between the four villages analyzed (93.1–99.3%).

The highest level of education was primary school in most households (82.3%), and 10.7% of households had no formal education. Only 7.1% of responding households had a family member with a secondary or higher level of education. Education levels varied among the four included villages, with Negezi and Ubata reporting higher levels of secondary or higher education (10.8% and 9.6%, respectively). The villages with the highest proportion of individuals without formal education were Ubata and Ngunga (21.3% and 14.3%, respectively).

### 3.2. Drinking Water Sources and Sanitation

Across all households, 270 (34.7%) reported obtaining drinking water exclusively from clean water sources. Similarly, 235 (30.2%) households used only clean water sources for other, non-drinking uses (see Table 1). For households listing a clean or improved water source as the primary source of drinking water, 70.7% used a covered or improved well as their primary source of drinking water, whereas 24.8% used dedicated filtered plumbing, 3.3% used the dedicated solar water treatment center, and 1.1% used a covered or improved spring. For households listing an unclean or unimproved water source as the primary source for drinking water, 40.1% obtained their water from untreated public water, 20.8% used open wells, 14.9% used open springs, and 19.4% used groundwater or other surface water sources (lakes, rivers, etc.; see Figure 3). Only one household reported using rain catchment as a primary source of drinking water, and no households reported using bottled water. Many households reported traveling long distances to procure water multiple times per week; the median trip rate among all households was 4 (3–7) trips per week, with 48.9% of households spending more than 2 h per trip.

The majority of households kept drinking water separate from other water storage at home (87.8%, see Figure 4). For water treatment, 334 (42.9%) households treated their water using conventional methods such as boiling, bleach/chlorine sterilization, cloth/mesh filtration, dedicated water filter use, or ultraviolet/sunlight sterilization. 

Regarding hygiene and sanitation, 448 (58.0%) of respondents reported washing hands with soap after urination or defecation, and 285 (37.4%) reported washing their hands with soap before eating or cooking. Just under half of households reported using protected methods for waste disposal, such as sewage systems and ventilated pit latrines (348, 44.7%), whereas others used ad hoc solutions. When asked specifically about child waste removal, 627 (80.5%) of responding households reported regular use of a toilet or latrine, while the remainder reported disposing of child waste in the garbage, burying it, or leaving it in the surrounding environment (see Table 1).

### 3.3. Prevalence and Treatment of Diarrhea in Under-Five Children

Among all households surveyed, 250 (32.1%) reported at least one child under five years of age with diarrhea over a two-week interval, and the total prevalence of diarrhea in under-five children was 25.6% (see Table 2). The median age of under-five children with diarrhea was 2 (1–4) years, and 50.4% of young children with diarrhea were female.

Among children with reported diarrhea quality, 9 (2.8%) had bloody stool and 15 (4.7%) had stool with visible ova or parasites. The majority of young children with diarrhea were treated for their illness; of those surveyed, 300 (87.5%) received medication for their diarrhea and 285 (83.1%) were brought to a local clinic for evaluation of their diarrhea. There was variability in treatments given for young children with diarrhea, with 132 (44.0%) receiving antibiotics, 53 (17.7%) receiving antiparasitics, 82 (27.3%) receiving oral rehydration solution (ORS), and 33 (11.0%) receiving traditional or other medicines.

### 3.4. Predictive Factors Leading to Diarrhea in Under-Five Children

On a per-household basis, multiple WASH-based factors were found to be significantly associated with decreased risk of diarrhea in children under five years of age in both univariable and multivariable analyses (see Table 3). In terms of water procurement, exclusive use of clean or improved water sources for drinking water, use of a dedicated transportation vessel for drinking water, living within two hours of a water source, and more frequent visits to obtain water were all associated with decreased risk of young childhood diarrhea in univariable analysis. Among these factors, use of a clean or improved drinking water source (adjusted odds ratio, aOR 0.060, 95% CI 0.42–0.84; *p* = 0.003) and decreased water procurement time (aOR 0.59, 95% CI 0.42–0.82; *p* = 0.002) were found to be independently predictive of decreased under-five diarrhea risk in multivariable analysis. Concerning water storage, storing drinking water in a dedicated separate container (OR 0.25, 95% CI 0.18–0.36; *p* < 0.001) and separation of drinking water from other water supplies (OR 0.38, 95% CI 0.24–0.59; *p* < 0.001) were similarly associated with decreased diarrhea risk, although these variables did not meet criteria to be included in the multivariable model. Regarding hygiene practices, handwashing with soap both after toilet use and before eating/cooking was associated with a protective effect against under-five diarrhea, although only handwashing with soap after toilet use was associated with significantly decreased under-five diarrhea risk in multivariable analysis (aOR 0.62, 95% CI 0.41–0.94; *p* = 0.024). Finally, sanitary practices were also found to be predictive of under-five childhood diarrhea between households, with use of improved sewage systems and disposal of child waste in designated toilets/latrines significantly associated with decreased risk of under-five diarrhea. Of these, use of improved sewage systems was found to be independently protective against under-five diarrhea in multivariable analysis as well (aOR 0.40, 95% CI 0.28–0.56; *p* < 0.001).

When considering aggregate prevalence of diarrhea among all under-five children, most factors found to be significantly protective against this outcome in the household analysis were also observed to be protective at the individual child level (see Table 3). Exceptions included using a clean source of drinking water (OR 0.83, 95% CI 0.65–1.05; *p* = 0.13) and washing hands with soap before cooking and eating (OR 0.85, 95% CI 0.57–1.08; *p* = 0.18). While both of these parameters were not significant in this analysis, a nonsignificant trend for a protective effect was indicated. Subsequent multivariable cluster analysis accounting for multiple children within each household demonstrated that living within two hours from a water source (OR 0.69, 95% CI 0.55–0.86; *p* = 0.001), washing hands after toilet use (aOR 0.66, 95% CI 0.54–0.82; *p* < 0.001), and using an improved sewage system (aOR 0.42, 95% CI 0.32–0.54; *p* < 0.001) were independently associated with decreased risk of diarrhea in young children.

Using exclusively clean water sources for non-drinking water, practicing point-of-use treatment of drinking water, and the volume of water returned per trip were not significantly correlated with decreased childhood diarrhea in either the per-household or per-child analyses. Similarly, the number of under-five children per household was not significantly associated with increased diarrhea risk in the per-household analysis (OR 1.14, 95% CI 0.96–1.34; *p* = 0.126).

## 4. Discussion

While early childhood diarrhea is decreasing in prevalence worldwide, it remains a significant issue in many developing countries. In this cross-sectional analysis, 32.1% of responding households in central Busiya reported having at least one child under five years of age suffering from diarrhea within a two-week period, exceeding national rates. As expected, the use of improved water sources, dedicated water containers, and WASH-based practices were found to be associated with decreased risk of diarrhea in young children; however, actual adoption of these practices was limited among households and variable between communities surveyed. These findings suggest multiple opportunities to improve childhood diarrhea rates in the rural Sub-Saharan African setting, with a particular focus on safe water storage techniques, regular handwashing, and improved sanitation to curb fecal–oral transmission.

### 4.1. Prevalence of Under-Five Diarrhea in Rural Tanzania

The aggregate two-week prevalence of diarrhea in children under age five in central Busiya was 25.6%, which is higher than reported national rates. In a 2019 post hoc analysis of the government-sponsored 2015–2016 Tanzania Demographic and Health Survey, the nationwide prevalence of under-five childhood diarrhea was calculated to be 12.1% (95% CI 11.3–12.9%) [34], and regional studies of rural Tanzanian communities using similar prevalence models have reported under-five childhood diarrhea rates between 6.1% and 16.9% [3,16]. Somewhat higher rates of under-five childhood diarrhea have been associated with urban Tanzanian environments rather than rural ones; for example, a 2018 cross-sectional study conducted in nearby Mwanza city reported an under-five diarrhea prevalence of 20.4% among urban communities [35]. With the exception of one village (Ubata, 50.0%), prevalence of childhood diarrhea in central Busiya generally fell between these calculated rural and urban rates, despite its rural setting (15.5–20.8%, see Table 2). This finding may reflect an increased need for safe water practices in Busiya at large, while highlighting the benefit of performing screening surveys to optimize resource allocation and determine areas most in need for targeted interventions.

### 4.2. Water Procurement Strategies for Preventing Under-Five Diarrhea

Improvement of local water sources has been a common intervention to increase quality of life and mitigate childhood diarrhea in low-income communities, and several studies have analyzed the potential benefits of these projects in Sub-Saharan Africa. In a 2018 Tanzanian cross-sectional microbial analysis of 97 well water samples in the rural Kilombero district, closed wells were found to have substantially lower levels of total coliforms, *Escherichia coli* counts, heterotrophic plate counts, and turbidity compared to open wells, although there was no significant difference between different types of closed wells [20]. Similarly, in a 2015 cluster-randomized controlled trial analyzing the direct effect of improving water sources in the rural Volta region in Ghana, drilling or rehabilitating boreholes independently reduced under-five diarrheal prevalence by 11% (95% CI 3–18%), with both control and intervention groups demonstrating comparable WASH compliance [36]. Despite this positive effect, the benefits of water source improvement can be negated by poor hygiene and sanitation after water procurement, and several studies have found simple water source improvements do not directly influence under-five diarrhea risk when considering other factors [34,37]. Accordingly, while procurement of drinking water from clean or improved sources was associated with decreased risk of young childhood diarrhea in this analysis, the protective effects were not substantial and the difference was only significant in the per-household analysis, not on a per-child basis (see Table 3).

In an attempt to provide an all-in-one solution for water source improvement, decentralized solar- or wind-powered water treatment centers placed at local water sources have become increasingly popular in recent years. These systems present an attractive option for low-income rural settings since they not only provide a self-contained and low-maintenance pump/treatment solution, but also promote incentives for income generation and microenterprise among local communities [2]. To date, long-term implementation data regarding these projects are sparse; however, standalone membrane filtration and kiosk-based water distribution systems have been independently vetted in similar cross-sectional studies [19,38]. In Busiya, adoption of the solar water treatment center at Nhobola was limited, with only select families reporting regular use of the facility. However, awareness of this available novel water treatment modality was significantly correlated with decreased risk of diarrhea at the per-child level (OR 0.73, 95% CI 0.55–0.97; *p* = 0.03), suggesting that further efforts to demystify and promote this resource may lower barriers and improve efficacy in coming years (see Table 3).

In addition to water source quality, transportation and storage of drinking water have also been identified as possible factors affecting young childhood diarrhea in developing countries. In a 2018 meta-analysis of 135 water and sanitation studies by Wolf et al. point-of-use filter interventions with safe storage were shown to reduce diarrhea risk by 61%, while at-home piped water of higher quality and continuous availability were shown to reduce under-five childhood diarrhea risk by 75% and 36%, respectively. Moreover, these interventions compared favorably to sanitation measures such as handwashing with soap and proper waste disposal (30% and 25% reduction, respectively) [39]. In this current analysis, just as sanitation-related interventions were among the strongest protective factors against under-five childhood diarrhea, so were decreased transportation burden and protected long-term storage of drinking water. In particular, use of designated storage containers for drinking water, short and frequent trips to procure water, and consistent separation of drinking water prior to consumption were all associated with a decreased risk of under-five diarrhea in both per-household and per-child analyses. These findings suggest that improper storage of drinking water is a major contributor to water contamination in central Busiya, with most contamination occurring at home when water is stored for extended periods of time.

### 4.3. Effects of Point-of-Use Water Treatment and WASH Practices on Under-Five Diarrhea

Despite widespread practice (42.9%, see Table 1), treatment of drinking water at home was not significantly associated with under-five childhood diarrhea in this study (see Table 3). This metric was inclusive of multiple treatment modalities, with boiling, cloth/mesh filtration, and dedicated filter use most implemented (see Figure 4). To date, multiple studies have demonstrated the efficacy of point-of-use drinking water treatment prior to consumption, with boiling as the most widely touted methodology [3,4,40]. In addition, non-boiling treatments, such as chlorination, filtration, and ultraviolet sterilization, have also been demonstrated to have a comparable positive effect on diarrheal outcomes in rural settings [41]. In this analysis, it is possible that treatment modalities were inadequately performed or other factors mitigated their effects. For instance, if water treatment occurred long before consumption, recontamination could potentially occur during periods of stagnation, an effect that could be exacerbated by prolonged storage times and poor on-site sanitation.

Like transportation and storage, proper WASH practices at home were also strong predictive factors of under-five childhood diarrhea in central Busiya. In both per-household and per-child analyses, effective sewage management and handwashing after latrine use were significantly associated with decreased risk of under-five childhood diarrhea, whereas handwashing before food preparation/consumption and proper disposal of child waste were only significantly correlated with decreased diarrhea in the household model (see Table 3). This connection between WASH adherence and improved diarrheal outcomes has been established in similar cross-sectional studies in both rural and urban environments, with most reporting some benefit from regular handwashing with soap and use of improved toilets such as pit latrines or dedicated sewage systems [3,16,42,43,44]. Moreover, handwashing with soap has been further demonstrated to independently reduce under-five diarrhea rates in midsized cluster-randomized controlled trials including several hundred households in the rural Sub-Saharan African setting [45,46].

Despite these findings, several studies have called into question the efficacy of WASH implementation in larger populations. In a 2018 cluster-randomized controlled trial of 8246 households and 6583 children in rural Kenya, WASH practices were found to have no correlation with diarrhea prevalence in children up to two years after birth, with modest improvements in childhood growth indicators seen only when paired with nutritional counseling [47]. Likewise, the 2019 cluster-randomized SHINE trial in rural Zimbabwe demonstrated a similar effect among 5280 households and 3686 children: over an 18-month follow-up of newborn infants, WASH practices were found to have no significant association with child length, hemoglobin concentrations, or diarrhea rates, whereas nutrition-based interventions were found to be protective against both stunting and anemia [48]. Further quantitative analysis of routinely collected stool samples in this population revealed that WASH practices significantly decreased the number of parasites detected in stool but had no effect on levels of bacteria or viruses [49]. These results suggest that WASH practices may play a smaller role in childhood diarrhea prevention when averaged on a larger scale, while other factors not considered in the present study, such as nutrition, may have a greater effect on early childhood health.

### 4.4. Influence of Social Factors on Under-Five Diarrhea

Other social factors have been analyzed for their potential effect on young childhood diarrhea rates. For example, the number of children per household has been found to correlate with increased risk of under-five childhood diarrhea in some cross-sectional studies, with the rationale that additional children might increase the likelihood of cross-contamination and may worsen sanitary conditions in the household [37]. Other analyses have only found increased diarrhea risk with very high numbers of children (i.e., >5 children per household) [50], and still others have found no significant correlation between multiple children and diarrhea risk [34]. Accordingly, in this analysis, the number of children per household was not significantly correlated with increased or decreased risk of under-five diarrhea (OR 1.14, 95% CI 0.96–1.34; *p* = 0.126). In other studies, education, occupation, and socioeconomic status of households have also been explored as potential influencing factors on young childhood diarrhea, with results varying substantially between study environments [2,3,16,34,37]. Given the relative homogeneity of these attributes across households in central Busiya, these factors were not explored in-depth in this analysis.

### 4.5. Treatment of Diarrhea in Under-Five Children

Treatment was widely available for children with diarrhea in central Busiya, with 87.5% receiving some form of therapy for their illness. Among those receiving treatment, the most popular modalities were antibiotics (44.0%), ORS (27.3%), and antiparasitics (17.7%), although traditional or other treatments were also employed (11.0%, see Table 2). These findings are partly a reflection of significant government intervention to reduce childhood diarrhea throughout Tanzania since the 1980s, with a strong focus on increasing availability of antidiarrheal treatment among underserved communities. In particular, multiple historical campaigns from the national Control of Diarrheal Disease (CDD) and the Integrated Management of Childhood Illness (IMCI) programs focused on the establishment of designated diarrhea treatment centers (DTCs) in primary health care facilities for specialized preparation of ORS, distribution of antibiotics, and administration of rotavirus vaccines [21]. While not specifically addressing prevention, these types of interventions have been essential in reducing childhood mortality from diarrhea, not only in Tanzania but also in other countries. In a 2019 comparative retrospective analysis of United Nations data, Black et al. determined that strategies involving direct diarrhea treatment via ORS, zinc, antibiotics, or rotavirus vaccines accounted for 49.7% of global diarrhea mortality reduction in children under 5 years old between 1980 and 2015. By comparison, interventions involving nutritional improvements or clean water practices accounted for 38.8% and 11.5% of this reduction, respectively [7].

### 4.6. Limitations

This study has several limitations. For one, this analysis was cross-sectional by design and reliant on voluntary, self-reported statistics from participating HOHs, rendering it subject to selection bias and recall/information bias. Selection bias was likely most influential in areas with lower response rates (e.g., Ngunga, 79.2%), which may have led to skewed reporting of under-five diarrhea rates in these communities, even though response rates in this study were high overall (98.5%). Similarly, recall bias likely had a significant effect on the results of this analysis, although the large number of participating households helped to mitigate this effect, and the directionality of this bias was indeterminate.

In reporting survey outcomes, multiple simultaneous water and sanitation practices among households were grouped and simplified for clarity of analysis, with potential disregard for temporality, regularity, and extent of practice, as well as seasonal changes. While these groupings were largely predicated on established WHO/UNICEF guidelines, the method by which various water and sanitation procedures were categorized as “clean” or “unclean” may have affected the perceived outcomes of this analysis. Furthermore, while two separate multivariable analyses were conducted to adjust for confounding factors in predicting under-five diarrhea, these analyses were not exhaustive, and many demographical, environmental, health-related, and individual factors may have influenced outcomes but were not included in these analyses. For included variables, within-model adjustments were made to account for additional potential confounding factors (e.g., the use of a GEE model in the “per-child” analysis to account for clustering of children within households), but these adjustments were not applied to all models universally due to lack of empiric efficiency in some scenarios. For example, households were not clustered by village due to high variation in water/sanitation variables irrespective of village diarrhea rates, leading to large standard errors for predictive variables and non-useful derived coefficient estimates. In these situations, simple multivariable models were used to provide a reasonably close estimation of increased/decreased under-five diarrhea risk for most WASH-related covariates of interest. Finally, this study was unable to capture and quantify sociocultural influences, including regional knowledge about clean water practices, clinic closures and logistics, and changes in local government policy.

Most importantly, Busiya and Shinyanga as a whole are rapidly developing, with planned expansion of electricity and water infrastructure in the upcoming years. These changes may significantly alter under-five diarrhea rates in these areas, improving quality of life measures for the people living in these communities in the near future.

## 5. Conclusions

In this cross-sectional analysis of rural communities in northwestern Tanzania, under-five early childhood diarrhea was found to have high prevalence, which may in part be due to limited adoption of clean water and sanitation practices in this population. However, multiple practices were found to have a significant protective effect against early childhood diarrhea in this environment, particularly those involving the minimization of water transportation and storage, as well as those promoting sanitary waste disposal and handwashing. Obtaining drinking water from improved sources was also associated with decreased diarrhea risk, albeit to a lesser extent. For children with diarrhea, treatment was generally available and consisted primarily of antibiotics and ORS. Together, these findings may inform potential targeted strategies for the reduction in young childhood diarrhea rates in the rural Sub-Saharan African setting. Further longitudinal studies are needed to identify social and logistical barriers preventing widespread adoption of clean water practices in these communities.

## Figures and Tables

**Figure 1 ijerph-19-04218-f001:**
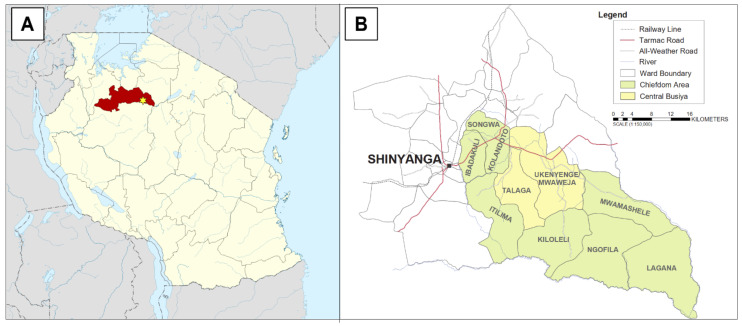
Study location. (**A**) Location of the Busiya chiefdom (star) within the eastern Shinyanga region (red) in northwestern Tanzania; (**B**) detailed map of wards included in the Busiya chiefdom, with central Busiya highlighted in yellow. Tanzania political map courtesy of Wikimedia Commons. Busiya regional map courtesy of the Busiya chiefdom traditional leadership.

**Figure 2 ijerph-19-04218-f002:**
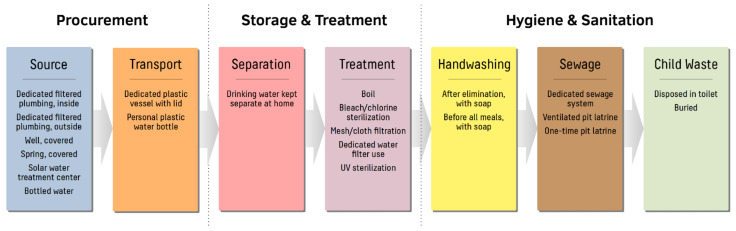
Definitions of clean water, sanitation, and hygiene (WASH) practices in central Busiya. Survey questions concerning WASH-related practices were categorized into water procurement, storage and treatment of drinking water, and at-home hygiene and sanitation. Delineations between clean/unclean practices were based on WHO/UNICEF guidelines with several site-specific modifications. WASH compliance was based on exclusive adherence to improved practices, without concurrent utilization of unimproved modalities.

**Figure 3 ijerph-19-04218-f003:**
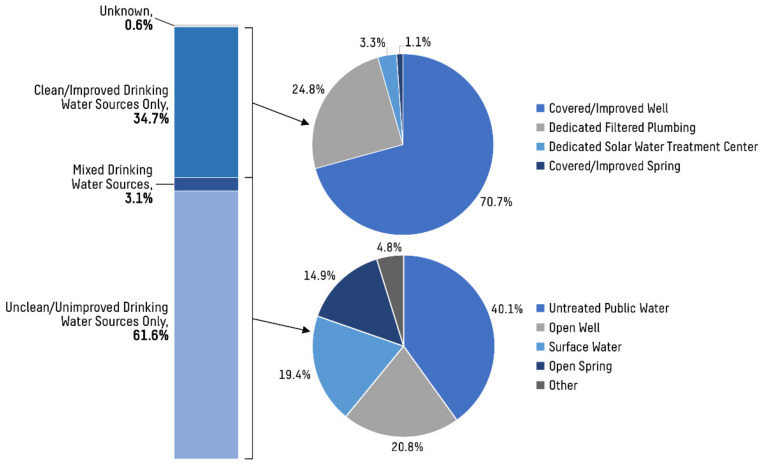
Distribution of primary drinking water sources for households in central Busiya. A minority of households obtained their drinking water exclusively from clean water sources (34.7%), and primary water sources among this group were largely homogenous (70.7% from covered or improved wells). Households obtaining their water by other means had more diverse procurement strategies, utilizing a wider variety of sources.

**Figure 4 ijerph-19-04218-f004:**
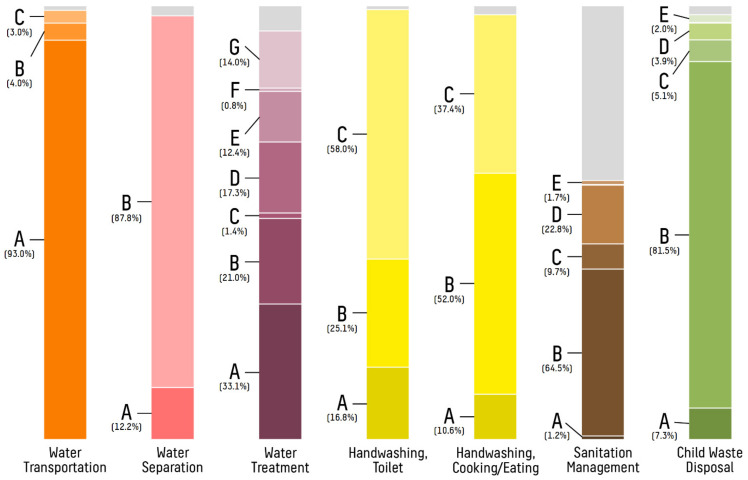
Distribution of primary practices for select WASH modalities among households in central Busiya. Reported primary WASH-based practices varied substantially between households. Non-primary practices are excluded, and unknowns are shown in gray. Water transportation: (A) plastic bucket with lid; (B) jerrycan; and (C) other. Water separation: (A) drinking water not separated; and (B) drinking water separated. Water treatment: (A) no treatment; (B) boiling; (C) bleach/chlorine treatment; (D) cloth/mesh filtration; (E) dedicated water filter; (F) ultraviolet sterilization; and (G) water settlement/skim. Handwashing, toilet: (A) none; (B) without soap; and (C) with soap. Handwashing, cooking/eating: (A) none; (B) without soap; and (C) with soap. Sanitation management: (A) sewage system; (B) ventilated pit latrine; (C) one-time pit latrine; (D) bucket; and (E) other. Child waste disposal: (A) in nature; (B) in toilet/latrine; (C) with garbage; (D) buried; and (E) other.

**Table 1 ijerph-19-04218-t001:** Household demographics and water, sanitation, and hygiene (WASH) practices.

Study Variable	Central Busiya (Total)	Nhobola	Negezi	Ngunga	Ubata
Total households, *n*	779	261	164	156	198
Total individuals, *n*	4702	1747	1077	1033	845
Individuals per household, median (IQR)	6 (4–7)	6 (5–8)	6 (4–8)	6 (5–8)	4 (3–5)
Employed as subsistence farmer, *n* (%)	729/761 (95.8%)	242/260 (93.1%)	157/162 (96.9%)	148/149 (99.3%)	182/190 (95.8%)
Highest level of education in the household					
None	77/723 (10.7%)	10/240 (4.2%)	6/148 (4.1%)	21/147 (14.3%)	40/188 (21.3%)
Primary	595/723 (82.3%)	219/240 (91.3%)	125/148 (84.5%)	122/147 (83.0%)	129/188 (68.6%)
Secondary	41/723 (5.7%)	4/240 (1.7%)	16/148 (10.8%)	3/147 (2.0%)	18/188 (9.6%)
Vocational/College	10/723 (1.4%)	7/240 (2.9%)	1/148 (0.7%)	1/147 (0.7%)	1/188 (0.5%)
Child demographics					
Total children, *n* (%)	1338/4702 (28.5%)	453/1747 (25.9%)	274/1077 (25.4%)	269/1033 (26.0%)	342/845 (40.5%)
Children per household, median (IQR)	2 (1–2)	2 (1–2)	2 (1–2)	1.5 (1–2)	2 (1–2)
Child age, median (IQR)	3 (2–4)	3 (2–4)	4 (2–4)	4 (2–4)	3 (2–4)
Number of females, *n* (%)	678/1334 (50.8%)	234/450 (52.0%)	144/274 (52.6%)	132/269 (49.1%)	168/341 (49.3%)
WASH compliance at household level					
Clean/improved source of drinking water, *n* (%)	270/779 (34.7%)	54/261 (20.7%)	45/164 (27.4%)	111/156 (71.2%)	60/198 (30.3%)
Clean/improved source of non-drinking water, *n* (%)	235/779 (30.2%)	99/261 (37.9%)	45/164 (27.4%)	52/156 (33.3%)	39/198 (19.7%)
Dedicated transportation vessel for drinking water, *n* (%)	648/771 (84.0%)	240/259 (92.7%)	118/162 (72.8%)	121/153 (79.1%)	169/197 (85.8%)
Dedicated transportation vessel for non-drinking water, *n* (%)	590/776 (76.0%)	193/261 (73.9%)	111/161 (68.9%)	121/156 (77.6%)	165/198 (83.3%)
Water procurement time <2 h, *n* (%)	396/775 (51.1%)	203/257 (79.0%)	85/164 (51.8%)	57/156 (36.5%)	51/198 (25.8%)
Times visiting water source per week, median (IQR) *	4 (3–7)	4 (3–6)	7 (1–7)	7 (6.5–21)	3 (2–5)
Amount of water returned per trip in liters, median (IQR) ^†^	100 (40–140)	120 (100–200)	40 (40–40)	100 (60–130)	120 (60–160)
Storage of water in a different location than transport, *n* (%)	574/753 (76.2%)	244/247 (98.8%)	97/158 (61.4%)	143/154 (92.9%)	90/194 (46.4%)
Separation of drinking water, *n* (%)	668/761 (87.8%)	240/258 (93.0%)	138/158 (87.3%)	153/155 (98.7%)	137/194 (70.6%)
Point-of-use drinking water treatment at home, *n* (%)	334/779 (42.9%)	88/261 (33.7%)	82/164 (50.0%)	114/156 (73.1%)	50/198 (25.3%)
Handwashing with soap after urination or defecation, *n* (%)	448/772 (58.0%)	138/258 (53.5%)	124/162 (76.5%)	109/155 (70.3%)	77/197 (39.1%)
Handwashing with soap before eating or cooking, *n* (%)	285/763 (37.4%)	69/250 (27.6%)	88/162 (54.3%)	55/154 (35.7%)	73/197 (37.1%)
Adequate sewage management at home, *n* (%)	348/779 (44.7%)	250/261 (95.8%)	63/164 (38.4%)	29/156 (18.6%)	6/198 (3.0%)
Adequate disposal of child waste, *n* (%)	627/779 (80.5%)	224/261 (85.8%)	114/164 (69.5%)	111/156 (71.2%)	178/198 (89.9%)

* Number missing = 136. ^†^ Number missing = 44. IQR = interquartile range (Q1–Q3).

**Table 2 ijerph-19-04218-t002:** Characteristics of under-five children with diarrhea.

Study Variable	Central Busiya (Total)	Nhobola	Negezi	Ngunga	Ubata
Households reporting a child with diarrhea, *n* (%)	250/779 (32.1%)	57/261 (21.8%)	40/164 (24.4%)	41/156 (26.3%)	112/198 (56.6%)
Diarrhea among children					
Frequency, *n* (%)	343/1338 (25.6%)	70/453 (15.5%)	57/274 (20.8%)	45/269 (16.7%)	171/342 (50.0%)
Child age, median (IQR)	2 (1–4)	1 (1–3)	2 (1–4)	3 (2–4)	2 (1–4)
Number of females, *n* (%)	173/343 (50.4%)	38/70 (54.2%)	31/57 (54.3%)	18/45 (40.0%)	86/171 (50.3%)
Quality of diarrhea among children					
Mucoid, *n* (%)	102/321 (31.8%)	12/68 (17.6%)	5/53 (9.4%)	13/44 (29.5%)	72/156 (46.2%)
Watery, *n* (%)	195/321 (60.7%)	47/68 (69.1%)	40/53 (70.2%)	26/44 (59.1%)	82/156 (52.6%)
Bloody, *n* (%)	9/321 (2.8%)	0/68 (0.0%)	7/53 (13.2%)	1/44 (2.3%)	1/156 (0.6%)
Ova and Parasites, *n* (%)	15/321 (4.7%)	9/68 (13.2%)	1/53 (1.9%)	4/44 (9.1%)	1/156 (0.6%)
Clinic visits for diarrhea, *n* (%)	285/343 (83.1%)	40/70 (57.1%)	42/57 (73.9%)	37/45 (82.2%)	166/171 (97.1%)
Treatment of diarrhea among children, *n* (%)	300/343 (87.5%)	45/70 (64.3%)	47/57 (82.5%)	41/45 (91.1%)	167/171 (97.7%)
Type of treatment for diarrhea among children					
Antibiotics, *n* (%)	132/300 (44.0%)	1/45 (2.2%)	22/47 (46.8%)	8/41 (19.5%)	101/167 (60.5%)
Antiparasitics, *n* (%)	53/300 (17.7%)	11/45 (24.4%)	7/47 (14.9%)	10/41 (24.4%)	25/167 (15.0%)
Oral rehydration solution, *n* (%)	82/300 (27.3%)	24/45 (53.3%)	7/47 (14.9%)	18/41 (43.9%)	33/167 (19.8%)
Traditional and other medicines, *n* (%)	33/300 (11.0%)	9/45 (20.0%)	11/47 (23.4%)	5/41 (12.2%)	8/167 (4.8%)

**Table 3 ijerph-19-04218-t003:** Predictors of diarrhea in under-five children, by household and by child (OR > 1 implies increased risk of diarrhea).

	HOUSEHOLD						CHILD					
	Univariable			Multivariable			Univariable			Multivariable		
Model Variable	OR	95% CI	*p*-Value ^§^	aOR	95% CI	*p*-Value ^§^	OR	95% CI	*p*-Value ^§^	aOR	95% CI	*p*-Value ^§^
Clean/improved source of drinking water	0.72	0.52–0.99	**0.04**	0.60	0.421–0.842	**0.0034**	0.83	0.65–1.05	0.13			
Clean/improved source of nondrinking water	0.76	0.55–1.07	0.12				0.78	0.61–1.00	0.05			
Dedicated transportation vessel for drinking water *	0.54	0.36–0.78	**0.0011**				0.51	0.37–0.68	**<0.0001**			
Dedicated transportation vessel for nondrinking water *	0.53	0.38–0.75	**0.0003**				0.61	0.46–0.80	**0.0004**			
Water procurement time <2 h	0.49	0.36–0.67	**<0.0001**	0.59	0.420–0.816	**0.0016**	0.54	0.43–0.68	**<0.0001**	0.69	0.55–0.86	**0.0012**
Times visiting water source per week ^†^	0.92	0.88–0.96	**0.0002**				0.94	0.92–0.96	**<0.0001**			
Volume of water returned per trip, liters	1.00	1.00–1.00	0.33				1.00	0.99–1.00	0.56			
Storage water in a different location than transport *	0.25	0.18–0.36	**<0.0001**				0.35	0.29–0.43	**<0.0001**			
Separation of drinking water *	0.38	0.24–0.59	**<0.0001**				0.56	0.44–0.71	**<0.0001**			
Point-of-use drinking water treatment at home	1.00	0.74–1.35	0.98				0.92	0.74–1.15	0.46			
Handwashing with soap after urination or defecation	0.55	0.41–0.75	**0.0002**	0.62	0.410–0.939	**0.0238**	0.66	0.53–0.82	**0.0002**	0.66	0.54–0.82	**0.0001**
Handwashing with soap before eating or cooking	0.68	0.49–0.93	**0.02**	0.81	0.519–1.247	0.331	0.85	0.57–1.08	0.18			
Adequate sewage management	0.37	0.27–0.51	**<0.0001**	0.40	0.281–0.561	**<0.0001**	0.38	0.30–0.49	**<0.0001**	0.42	0.32–0.54	**0.0001**
Adequate disposal of child waste *	0.67	0.46–0.96	**0.03**				0.86	0.67–1.10	0.24			
Solar water treatment center knowledge ^‡^	0.71	0.50–1.02	0.06				0.73	0.55–0.97	**0.03**			

* Not included in multivariable analyses due to response predominance (>70%). ^†^ Not included in multivariable analyses due to missingness (>10%). ^‡^ Not included in multivariable analyses due to low relevance. **^§^** Statistically significant *p*-values are in boldface. OR = odds ratio; CI = confidence interval; aOR = adjusted odds ratio.

## Data Availability

The data presented in this study are available on request from the corresponding author. The data are not publicly available due to privacy considerations for enrolled subjects. Please contact P.H.M. via email: paulhmcclelland@protonmail.com.

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
