# Peer review of "Improved Water and Waste Management Practices Reduce Diarrhea Risk in Children under Age Five in Rural Tanzania: A Community-Based, Cross-Sectional Analysis"

_ijerph, 2022, doi:10.3390/ijerph19074218_

Round 1

Reviewer 1 Report

This manuscript is well written and a lot of effort has been invested in its preparation, including in the tables and figures. See my suggestions below: 

Methods:

-line 168, it would be helpful if the authors could make it clearer what the standards or cutoffs for compliance were. Perhaps those could be worked into Figure 2? For example, were they yes/no?

-lines 175-177, I suggest that the authors include more information about the nature of these items. For example, were they open-ended? If not, what were the possible response categories?

-line 181, please provide more information about the US-TZ research team. Were they students? What was their level of training and preparation?

-Please provide details about the sampling approach and recruitment.

-Please provide details about the control variables in the multivariate models and a justification for why they were included.

-There seem to be sizable differences among the study locations with respect to many of the important dependent variables. It may be worth considering some type of cluster analysis and if not, why not?

Results: 

-Figure 4: I can see that the authors are trying to communicate a lot of information in one single figure. It probably makes sense to them. As a reader, it was not immediately clear to me. I am not sure what I would do to change it, but the percentages included in the footnote sort of get in the way. Perhaps remove them?

-line 294, I don’t recall in the method section there being a description of these more clinical variables. How were they derived? Chart reviews or reports from HOH? Presumably on for those that had received care from a facility. More description in the Method section about these variables would be helpful.

-subheading 3.3: it seems a little disorganized to say ‘prevalence of diarrhea…’ then also include treatment variables in the same paragraph.

-3.4: I don’t personally feel there is much value added in including univariate results, especially if they’re just followed by multivariate results. In short, what’s the point? I feel it makes the results section unnecessarily long and complicated.

Reviewer 2 Report

This study was a household-based, cross-sectional survey of clean water, sanitation and hygiene practices and the prevalence of diarrhoea in children under five years of age in rural Tanzania. The organizational framework and resource allocation were based on local census data, which included demographic information and household statistics, which were then used to locate all households with children under the age of 5, which were included in the analysis using a questionnaire. My comments are as follows: 
1. It is better for the author to point out the innovative comparison of his own research with previous research.  
2. It is recommended to improve the graphic quality of the article.  
3. When conducting a cross-sectional study, in addition to the etiology reported in the literature, these etiology indicators should be stable, that is, they should not change with the occurrence of the outcome. 
4. It is recommended to explore multi-factors analysis, and the existing evidence, time sequence, and stability of indicators should be considered at the same time. 
5. In a causal analysis of the findings, it is suggested that the quality of water sources during different periods should be considered before emphasizing the positive effects of improved water sources and storage habits, but this does not appear to be reflected in the article. 
6. The cross-sectional research data itself has many biases, so it is recommended that the authors clarify the main sources of bias in each method in the article, whether the expected trends of all the main sources of potential bias are consistent, and whether the directional conclusions are consistent.
Overall, the research subjects and research methods of this study are sufficiently innovative, the case studies are sufficiently convincing, and the references are sufficient, timely and academic to provide insights into improving water and waste management practices in rural Tanzania. It has theoretical support and also helps reduce the risk of diarrhea in children under 5 years of age.

Round 2

Reviewer 2 Report

My comments have been successfully addressed.